# Divergent Effects of Factors on Crash Severity under Autonomous and Conventional Driving Modes Using a Hierarchical Bayesian Approach

**DOI:** 10.3390/ijerph191811358

**Published:** 2022-09-09

**Authors:** Weixi Ren, Bo Yu, Yuren Chen, Kun Gao

**Affiliations:** 1Key Laboratory of Road and Traffic Engineering of the Ministry of Education, College of Transportation Engineering, Tongji University, 4800 Cao’an Highway, Shanghai 201804, China; 2Engineering Research Center of Road Traffic Safety and Environment, Ministry of Education, Tongji University, 4800 Cao’an Highway, Shanghai 201800, China; 3Department of Architecture and Civil Engineering, Chalmers University of Technology, SE-412 96 Gothenburg, Sweden

**Keywords:** crash severity, autonomous driving, conventional driving, hierarchical Bayesian approach

## Abstract

Influencing factors on crash severity involved with autonomous vehicles (AVs) have been paid increasing attention. However, there is a lack of comparative analyses of those factors between AVs and human-driven vehicles. To fill this research gap, the study aims to explore the divergent effects of factors on crash severity under autonomous and conventional (i.e., human-driven) driving modes. This study obtained 180 publicly available autonomous vehicle crash data, and 39 explanatory variables were extracted from three categories, including environment, roads, and vehicles. Then, a hierarchical Bayesian approach was applied to analyze the impacting factors on crash severity (i.e., injury or no injury) under both driving modes with considering unobserved heterogeneities. The results showed that some influencing factors affected both driving modes, but their degrees were different. For example, daily visitors’ flowrate had a greater impact on the crash severity under the conventional driving mode. More influencing factors only had significant impacts on one of the driving modes. For example, in the autonomous driving mode, mixed land use increased the severity of crashes, while daytime had the opposite effects. This study could contribute to specifying more appropriate policies to reduce the crash severity of both autonomous and human-driven vehicles especially in mixed traffic conditions.

## 1. Introduction

Autonomous driving has recently been an innovation hotspot in the global automotive industry [1,2]. It is widely believed that full automation driving will be an important direction in the development of transportation engineering and will provide a potential solution to transportation-related issues in safety, efficiency, and mobility [3,4,5,6]. However, due to the incompleteness and cost of the current technological progress, the perception, identification, and decision-making systems of autonomous vehicles (AVs) are, as it stands, not perfect. They cannot effectively deal with all kinds of factors that affect driving safety. Heretofore, there have been hundreds of crashes with the autopilot system turned on, leading to heavy personal and property losses and psychological roadblocks to adopting AVs [7,8,9]. Impacting factors on the crash severity of AVs may be different from human-driven vehicles since autonomous driving is integrated and systematically based on big data and artificial intelligence, while conventional driving is personalized [10]. Additionally, there are a series of new problems brought about by AVs. For example, when driving autonomous vehicles on roadways, drivers may use their travel time to accomplish leisure activities which will inhibit their anticipation of possible driving activities and eventually result in a volatile traffic environment [11,12,13]. Therefore, it is meaningful and imperative to explore the divergent effects of factors affecting safety for autonomous and conventional driving using crash data.

The crash data involved with AVs has become more and more available for the public, due to the fact that the regulatory requirements for the development and testing of AVs have been gradually relaxed. AVs were allowed to be tested on roadways in September of 2014 [14]. Companies and manufacturers that were approved to test AVs on California public roads must submit a Traffic Collision Involving an Autonomous Vehicle Report (OL 316) of the full description of the collision and other valid information [15].

Crash analyses of AVs have arisen within the past years. Some studies have focused on the factors contributing to AV crash severity levels. The positive association between travel speed and crash severity has been widely reported [16]. A substantially higher likelihood of AV-involved injury crashes at the intersections was found [17,18]. The lengthy time for drivers to repossess the authority of the vehicle might increase the likelihood of serious incidents [19]. Down-slopes, nighttime, involvement of multiple vehicles, and high-density traffic would also increase the likelihood of high crash severity of AVs [20]. In addition, location at an intersection, presence of roadside parking has been found to be the main positive contributing factors to the severity of AV crashes, while the one-way road would decrease the crash severity [21].

A lot of research on disengagements has been conducted based on Disengagement Reports (OL311R) from the California Department of Motor Vehicles (DMV). Causes and contributing factors of disengagements were investigated, and when lacking certain numbers of radar and LiDAR sensors installed on AVs were found to significantly induce an AV disengagement [22]. Additionally, influence factors between disengagements with a crash, disengagements with no crash, and no disengagement with a crash in a mixed traffic environment were also discussed. Variables related to AV systems (such as software failures) and other roadway participants may increase the propensity of disengagement without a crash [23]. With the analysis of AV’s interactions with other road users before a collision in a temporal manner, the results showed that the most representative pattern in AV crashes was “collision following AV stop” [24]. Distinct from these previous studies, this study will not analyze disengagements separately but consider disengagement-related variables as additional explanatory variables to obtain more insights into the pre-crash behavior of Avs; then, distinct effects of factors on crashes between autonomous and conventional driving modes will be further explored in this study.

To analyze the influencing factors of crashes, many methods have been employed in previous studies, such as the probit model, binomial/multinomial logistic regression, classification tree, and so on [21,25,26]. However, reliable and unbiased correlations between crashes and influencing factors cannot be established because of the presence of unobserved heterogeneity [27,28,29]. The hierarchical Bayesian approach can solve such problems. In addition, since AV crash data are difficult to collect but gradually available, the hierarchical Bayesian approach can use any engineering experiences or justified previous findings as prior knowledge to update the model [30,31]. This approach could also well handle missing data that occur commonly in crash records by considering the information contained in other observed data [32,33]. In addition, such a technique performs well in the estimation of discrete outcome models with smaller sample sizes [34]. For example, this method was applied to analyze correlations of influencing factors of AV-involved crashes, with a sample of 113 available crashes [35].

Given the above, there has been much research on the influencing factors of crashes and disengagements involved with autonomous driving, but there is a lack of comparative analyses of those factors between AVs and human-driven vehicles. To fill this research gap, this study utilizes the publicly available AV crash data in real driving environments and employs the hierarchical Bayesian approach to further explore and examine the differences of impacting factors between autonomous and conventional driving modes from the aspect of crash severity (i.e., injury or no injury). This study will assist the understanding, development, and testing of autonomous driving systems. In the stage of human-machine co-driving, it can also provide reliable references for reducing crash severity for both AVs and human-driven vehicles.

## 2. Materials and Methods

### 2.1. Data Preparation

As mentioned before, when AVs were involved in a crash while driving on public roads in California, a description of how the collision occurred and other associated factors would be submitted in the Traffic Collision Involving an Autonomous Vehicle Report (OL 316). AVs in the conventionally human-driven mode still need to submit crash reports, therefore, crashes in both driving modes (i.e., the autonomous driving mode and conventional driving mode) were included in this database. These publicly available reports can be downloaded from the website (https://www.dmv.ca.gov/portal/vehicle-industry-services/autonomous-vehicles/autonomous-vehicle-collision-reports/ (accessed on 1 April 2021)). By the time of writing this paper, reports from May 2018 to March 2021 were open, fully informative, and available. The Society of Automotive Engineers (SAE) defines six levels of driving automation to describe the full range of driving automation features, from Level 0 (No automation) to Level 5 (full automation) [36]. Vehicles in this AV crash database are considered to be conditional automation (Level 3), also known as driver-initiated automation [35,37]. Level 3 AVs are equipped with the ADAS technologies, sensors, and actuators, capable of automated highway driving, automated city driving, automated valet parking, and evasive maneuvers, but it is still essential for test drivers to take over driving promptly if there is a foreseen crash [38].

To consider the transitions from AV systems to test drivers, more information from Disengagement Reports (OL 311R) provided by California DMV (https://www.dmv.ca.gov/portal/dmv/detail/vr/autonomous/testing (accessed on 1 April 2021)) was added, and these reports consisted of all instances of disengagements occurring when AVs were tested. Then, the disengagement-related data was linked with the AV crash database (i.e., Traffic Collision Involving an Autonomous Vehicle Reports (OL 316)). Since not all disengagements led to a crash, information from Disengagement Reports (OL311R) were matched to the AV crashes that involved disengagements by carefully comparing the specific dates, manufacturer, and vehicle types. In addition, a few crashes involving disengagements which could not be found in Disengagement Reports (OL 311R) but recorded by the descriptions of crashes in OL 316 were manually marked as “the presence of disengagement”. In this study, the autonomous driving mode includes two situations: (1) the AV system remained engaged throughout the crash; (2) the driver took over the AV before the crash (i.e., disengagement occurred). Conventional mode indicates that the manual mode is employed before the crash for a considerable period and the human driver independently responds to the crash. Crashes in the conventional driving mode were filtered by two criteria: (1) it was emphasized in OL 316 that the vehicle was manually driven before the crash and disengagement was not mentioned; (2) the crash cannot be found in the Disengagement Reports (OL311R). A total of 180 crashes in San Francisco were extracted and used in the final analysis, including 96 crashes in the autonomous driving mode and another 84 in the conventional driving mode. Crashes with disengagement accounted for about 35% in autonomous driving mode.

Six new variables (i.e., disengagement, initiator of disengagement, unwanted behavior of other roadway participants, unwanted movement of AVs, changing lane, deceleration) from Disengagement Reports (OL311R) were fully contained in the dataset for further analyses. Specifically, disengagement reflects the presence or absence of disengagement in the autonomous driving mode. The initiator of disengagement indicates whether the disengagement is initiated by the system or the test driver. The other four variables were extracted from the description of Disengagement Reports (OL311R), which may be the causes of disengagements. Unwanted behavior of other roadway participants means reckless action of another vehicle or another non-vehicle roadway participant, such as a cyclist driving aggressively. Illegal behavior of AVs, such as entering the opposite lane suddenly, is reflected by the unwanted movement of AVs. Changing lanes indicate lane-changing maneuvers of AVs for reasons such as unstable target lane model. Deceleration refers to AVs dropping the speed for safety precaution or other reasons. As for those disengagements only found in the crash reports of OL316, their disengagement-related variables were manually extracted based on the descriptions of crashes.

To better understand the impact of environmental, road, and vehicle characteristics on the safety under different driving modes, this study obtained more explanatory variables through TransBASE: Linking Transportation Systems to Our Health (http://transbasesf.org/transbase/ (accessed on 1 April 2021)) and Google Earth (https://www.google.com/earth (accessed on 1 April 2021)) and then made hard efforts to manually link them to the crash sites through the location of each crash. TransBASE is a free and open online database that currently includes over 200 spatially referenced variables from multiple agencies and across a range of geographic scales, including infrastructure, transportation, zoning, sociodemographic, and collision data, all linked to an intersection or street segment. It is currently used by San Francisco Municipal Transportation Agency. Seven environmental variables (metro stop, land use, muni line, daily visitors’ flowrate (DVF), pavement markings conditions, schools, parks) and 13 road variables (street classification, one-way, divided median, marked centerline, bike lane, on-street parking, off-street parking, traffic calming, sidewalk, driveway, crash lanes, speed limit, slope) were obtained from TransBase. Google Earth was used to supplement some information, such as the specific width of the road.

Crash severity was chosen to be the dependent variable. Considering the small amount of data, it was divided into two levels. Crashes with injuries were considered more serious, while a crash without injury (i.e., property damage only) was thought to be of lower severity. The research on crashes with injuries or not had unique implications, especially for autonomous vehicles. The public had great concerns about the safety of the AVs and crashes with injury or death of people have proven to be a potential deterrent to the acceptance and credibility of AVs. In addition, insurance, legal, ethical, economic, and other fields were also interested in whether there was an injury in a crash involved with AVs.

Before modeling, some typical variables that describe driving conditions were picked out and a percent-stacking bar chart was plotted. As can be seen from Figure 1, the proportion of each selected variable, such as the time of crashes (Night), road characteristics (Speed limit, Street width, Number of driveways, Street type), type of the crash places (Daily visitors’ flowrate (DVF), Intersection, Land use), and vehicle state at the time of crashes (Turning movement, Vehicle state) was similar, which meant crashes in the autonomous driving mode and conventional driving mode occurred under similar conditions, and they were comparable.

Various discrete and continuous variables were obtained. The continuous variables and their descriptive statistics are provided in Table 1. Before the model establishment, this study divided continuous variables into discrete variables. Variables, including the count of public and private schools within a quarter-mile, the count of parks within a quarter-mile, and the count of driveways along the segment, were divided into two groups, according to whether the number was less than 4. The rest continuous variables were split into two groups, such as the number of lanes at the crash site (great than 2 or not), the width of the street in feet (more than 60 feet or not), the speed limit (more than 25 mph or not), the slope of the road (larger than 3% or not), etc.

After discretization of the continuous variables, 41 discrete variables used in the model were finally obtained. Table 2 presents the dependent variable and the other 40 explanatory variables which are divided into three categories, including environment, roads, and vehicles. Specifically, there were 14 environmental variables (e.g., trees, land use, weather, roadway surface, etc.), 14 road variables (e.g., bike lanes, street width, number of driveways, etc.), and 12 vehicle variables, (e.g., vehicle damage, turning movement, manufacturer, vehicle year, vehicle state, etc.). The detailed descriptions, distributions, and sources of them are provided.

Before modeling, multicollinearity was checked by calculating the variance inflation factors (VIFs) for all independent variables. VIF indicates the extent to which an indicator’s variance is captured by the remaining indicators of a given construct and VIF > 10 denotes severe multicollinearity [39,40]. In this study, the VIF values for all the selected independent variables were less than 10, indicating that the problem of multicollinearity did not exist or could be negligible while modeling.

### 2.2. Hierarchical Bayesian Approach

This study applied the hierarchical Bayesian approach to explore the differences of impacting factors on crashes for both autonomous and conventional driving modes while considering the unobserved heterogeneities caused by vehicle companies and vehicle years. This approach was composed of two parts: the hierarchical model and the Bayesian inference.

#### 2.2.1. Hierarchical Model

The hierarchical method can properly model the potential heterogeneities [41,42,43], so the crash effects of explanatory variables can be analyzed more accurately by using this multi-level structure. In this part, the hierarchical logistic regression model was used to analyze the impact of different influencing factors on the crash severity. In particular, a “Vehicle company & year” unit was considered as a cluster, and there were several sub-clusters per cluster, i.e., each crash.

Previous studies show that taking vehicle units as observation units may reveal crash propensity variation among different vehicles [44,45]. Recently, more companies have become permit holders to test their AVs on roadways. The perception recognition system, decision-making system, software algorithm, and computing ability of AVs produced by different companies have a lot of differences [26]. It should also be noted that autonomous driving technologies are persistently and rapidly advancing, and the vehicle year can reflect the “older” or “newer” technology to a certain extent. The complex influence of such unobserved factors on the correlation between other observed variables and dependent variables, called unobserved heterogeneity, may result in biased indications. These variables cannot be obtained, but they could be reflected by AVs’ company and production year to a certain degree. Therefore, this study took vehicles with the same vehicle year from the same company (i.e., the “Vehicle company & year” unit) as an observation unit to alleviate the effects of unobserved heterogeneity.

In the analysis of crash severity, the response variable $Y_{ij}$ for the ith crash in the jth vehicle unit is a dichotomous variable, such that $Y_{ij}$ = 1 means high severity (i.e., injury crash), while $Y_{ij}$ = 0 represents low severity (i.e., no injury crash). The likelihood of $Y_{ij}$ = 1 is denoted by $\pi_{ij}=\mathrm{Pr}\left(Y_{ij}=1\right)$ which follows a binomial distribution. In level 1 (crash level), the likelihood of $Y_{ij}$ = 1 is described as follows:(1)$$\log it(\pi_{ij})=\log\left(\frac{\pi_{ij}}{1-\pi_{ij}}\right)=\beta_{0j}+\overset{P}{\underset{p=1}{\sum }}\beta_{pj}X_{pij}+\varepsilon_{ij}$$
where $\beta_{0j}$ is the level 1 intercept; $\beta_{pj}$ is the regression coefficient for $X_{pij}$; $X_{pij}$ is the value of the pth independent variable for crash i for vehicle unit j; P is the number of independent variables in level1; $\varepsilon_{ij}$ is the disturbance term with mean zero and variance to be estimated.

In the context of the hierarchical model, the within-crash correlation is specified in the “Vehicle company & year” level (level 2) as:(2)$$\beta_{0j}=\gamma_{00}+\overset{Q}{\underset{q=1}{\sum }}\gamma_{0q}Z_{qj}+\mu_{0j}$$
(3)$$\beta_{pj}=\gamma_{p0}+\mu_{pj}$$
where $\gamma_{00}$ and $\gamma_{p0}$ are estimated intercepts in the “Vehicle company & year”-unit level; $Z_{qj}$ is the qth independent variable for “Vehicle company & year”-unit j; $\mu_{0j}$ and $\mu_{pj}$ are the random effects varying across “Vehicle company & year”-units for the crash-level intercept and covariate p, and they are assumed as normal distributions with means zero and variances $\sigma_{0}^{2}$ and $\sigma_{k}^{2}$, respectively.

Both $\beta_{0j}$ and $\beta_{pj}$ vary with the different “Vehicle company & year” units, in which two components are combined to decide the coefficient values. First, it’s assumed that they have linear relationships with the level 2 covariates $Z_{qj}$, because various environmental, road, and vehicle features may result in different severity levels. Second, besides the fixed parts which depend on the level 2 covariates $Z_{qj}$, random effects are also included ($\mu_{0j}$ and $\mu_{pj}$). The random effects between “Vehicle company & year” units only vary across the different units, but in the same unit, they are constant for the crash. Previous studies have shown that considering these random effects, potential random variations across “Vehicle company & year” units are allowed and correlations within them can be explained [46,47].

The full model with Equation (4) is the hierarchical model with both random intercept and random slope [30].
(4)$$\log it(\pi_{ij})=\log\left(\frac{\pi_{ij}}{1-\pi_{ij}}\right)=\gamma_{00}+\overset{Q}{\underset{q=1}{\sum }}\gamma_{0q}Z_{qj}+\mu_{0j}+\overset{P}{\underset{p=1}{\sum }}\gamma_{p0}X_{pij}+\overset{P}{\underset{p=1}{\sum }}\mu_{pj}X_{pij}+\varepsilon_{ij}\gamma_{00}+\overset{Q}{\underset{q=1}{\sum }}\gamma_{0q}Z_{qj}+\mu_{0j}+\overset{P}{\underset{p=1}{\sum }}\gamma_{p0}X_{pij}+\overset{P}{\underset{p=1}{\sum }}\mu_{pj}X_{pij}+\varepsilon_{ij}$$

#### 2.2.2. Bayesian Inference

To calibrate the hierarchical model, this study employed the Bayesian inference technique. The Bayesian inference technique is a prevailing way to explicitly model the hierarchical structure in which prior beliefs and the likelihood function of data at hand are fused to obtain the marginal posterior distribution. The distinctions between fixed and random effects disappear since all effects are now considered to be random and the hierarchical structure is accounted for. Compared to classical frequentist methods, Bayesian inference shows a lot of theoretical and practical advantages in road safety analysis, such as the good capability to handle small size data and deal with missing data commonly occurred in crash records, allowing for a comparison of any number of non-nested models, and considering hierarchies in the model [48,49,50].

Prior distribution is a material part of Bayesian inference, and the following three kinds of prior distribution are commonly used [51]: (a) strong informative prior distributions based on expert knowledge or previous investigation; (b) weak informative prior distributions that do not dictate the posterior distribution significantly but are able to prevent inappropriate inferences; (c) uniform priors that could interpret evidence from the data probabilistically [34,35,45,52,53]. In the absence of prior information, uniform priors were used in this study. For the regression coefficients ($\gamma_{00}$, $\;\gamma_{0q}$, and $\gamma_{p0}$), normal distributions (0, 1000) were assumed. A study of Fink [54] showed that the conditional conjugacy property of inverse-gamma priors suggested more flexible mathematical properties, so the variance $\sigma_{0}^{2}$ and $\sigma_{k}^{2}$ were assumed to be distributed as Gamma (0.001, 0.001). As previously mentioned in Section 2.2.1, $\sigma_{0}^{2}$ and $\sigma_{k}^{2}$ are the variances of $\mu_{0j}$ and $\mu_{pj}$, respectively, and $\mu_{0j}$ and $\mu_{pj}$ are the random effects varying across “Vehicle company & year”-units for the crash-level intercept and covariate.

A well-known computing approach for Bayesian inference, the Markov chain Monte Carlo (MCMC) method [55], was used in this study to better approximate the target posterior distribution. Two parallel MCMC chains were initiated for each model, and 5000 starting iterations in each chain were dropped as burn-in, whereas 10,000 iterations in each chain were used for generating the descriptive statistics for posterior estimates. In summary, the posterior estimates were based on 20,000 MCMC iterations (10,000 in each of the chains). The MCMC chains were reasonably converged since the ratio of pooled- and within-chain interval widths were around 1 [56].

To better interpret the results of the hierarchical Bayesian models, the odds ratios were calculated (Odds ratio = $e^{\gamma }$) to show relative likelihoods. For example, in the analysis of crash severity, with the independent variable switching from 0 to 1, the odds of high severity crash increases/decreases by a value of |$e^{\gamma }-1$|. This study used the 95% Bayesian credible interval (95% BCI) to examine the significance of variables. The coefficient estimations were identified to be significant if the 95% BCIs didn’t cover “0” or the 95% BCIs of odds ratio didn’t cover “1”.

In this study, Watanabe-Akaike Information Criterion (WAIC) [57] and Leave-one-out cross-validation (LOO) [58] were used to measure the model performance and select the best fitting model. LOO and WAIC have various advantages over simpler estimates of predictive error. WAIC can be viewed as an improvement on the deviance information criterion (DIC). It has been known that DIC has some problems for Bayesian models, which arises in part from not being fully Bayesian where DIC is based on a point estimate [59]. Distinct from DIC, WAIC is fully Bayesian in that it uses the entire posterior distribution, and it is asymptotically equal to Bayesian cross-validation. However, the study of Vehtari et al. [60] showed that LOO was recommended to be tried in the finite case with influential observations. Thus, in our study, the best-fitting models were those with the lowest WAIC and LOO values.

The hierarchical Bayesian approach was performed to model the binary outcome using the “brm” package in the statistical software R (version 3.4.4).

## 3. Results

As shown in Figure 2, in the conventional driving mode, the proportion of crashes with injury was 18%, similar to that of the autonomous driving mode (22%).

The best-fitting hierarchical Bayesian models with the lowest WAIC and LOO for crash severity in both driving modes were finally selected. The model for the autonomous mode included a total of eight explanatory variables, and that for the conventional mode contained seven explanatory variables. All these included variables were statistically significant and removing any of them would reduce the systemic utility of these models. Table 3 and Table 4 presents the results of the two hierarchical Bayesian models. To represent the data more intuitively, odds ratios (OR) are plotted in Figure 3 and Figure 4.

For the autonomous mode, raining presence, mixed land use, muni line presence, bike lanes presence, two sidewalks presence, and moving vehicle state were all positively associated with the crash severity, whereas daytime, and daily visitors’ flowrate (DVF) less than 3418 person-times had negative effects on crash severity.

For the conventional mode, the number of lanes at the crash site, bike lanes presence, turning movement presence, moving, and vehicle state all positively affected crash severity, whereas DVFs (<3418, 3418~11,982, and 11,982~40,040 person-times) were negatively associated with crash severity.

Detailed explanations of these influencing variables are provided below from the following three aspects. Additionally, Figure 5 demonstrates the comparison of the same influencing factors for injury crash propensity in the autonomous driving mode and conventional driving mode. In Figure 5, the posterior distribution of each influencing factor’s OR is plotted.

## 4. Discussion

### 4.1. Impacting Factors on Crash Severity

#### 4.1.1. Environmental Variables

The results revealed a positive, statistically significant correlation between the population flow size and crash severity in both driving modes. Compared to the maximum level of DVF (>11,982 person-times), the likelihood of high severity decreased in the other three levels in the conventional mode by 64%, 62%, and 59%, respectively. (DVF < 3418 person-times, OR = 0.36; DVF 3418~11,982 person-times, OR = 0.38; DVF 11,982~40,040 person-times, OR = 0.41) (See Table 4 and Figure 4). As shown in Table 3 and Figure 3, when compared to the maximum level of DVF, there was a decrease of 15% in the likelihood of high crash severity if DVF was less than 3418 person-times in the autonomous driving mode (OR = 0.85). The effect of other levels of DVF in this mode was not statistically significant. Figure 5a showed the different impacts of the minimum level of DVF on injury crash propensity in different driving modes.

As the results showed, injury crashes were more likely to occur in areas with higher DVF. There are often great safety hazards in densely populated areas such as bus stations, schools, and residential areas [61]. For the conventional driving mode, passing through these areas, nervousness, and anxiety of drivers may increase, which would affect the driving behavior [62]. For the autonomous driving mode, crowded areas would significantly increase the conflict points between vehicles, other transportation, and pedestrians, and the perceived ability of AVs may be influenced. However, AVs can reduce crash severity by avoiding driver errors (e.g., speeding, fatigue driving, aggressive driving, distracted driving, slow reaction times, etc.). Compared with humans, AVs are supposed to have better deceleration performance, leading to low collision speed in crowed sites (with higher DVF). Therefore, the impact of DVF on crash severity in the autonomous driving mode is smaller than that in the conventional driving mode.

Compared with the conventional driving mode, several other factors also significantly affected the likelihood of high crash severity in the autonomous mode (See Table 3 and Figure 3). The presence of muni lines increased the likelihood of high crash severity by 48% (OR = 1.48). Raining increased the likelihood of high crash severity in the crash by 9% (OR = 1.09). Compared to residential land use, there was a 19% growth in the possibility of high crash severity in mixed land-use types (OR = 1.19). Relative to the nighttime, daytime would reduce the likelihood of high crash severity in the crash by 21% (OR = 0.79).

The existence of the muni line may lead to a more complicated traffic environment in which buses, conventional vehicles, and AVs are mixed. As mentioned before, AVs may lead to some problems in the mixed flow. Previous studies suggest that AVs should be combined with Public Transportation (PT) systems to reduce labor costs, expand service hours and optimize the spatial and temporal allocation of the PT services [63,64]. However, the safety issues caused by the combination of AVs and PT should not be ignored in the current stage of AVs. In addition, mixed land-use patterns typically exhibit diverse land-use types leading to complex roadway layouts, line of sight occlusion, and other problems. A study also found that AVs may have critical issues in the case of complex environments [65]. Human drivers have a certain understanding of the general structure of each kind of thing on the road, and they can rapidly imagine the shape of the occluded object and deal with the occlusion problems well. However, for AVs, this is a problem that may lead to untimely braking. At present, there are still problems in the total factor recognition and perception of current autonomous driving vehicles in complex environments, one of which is that autonomous driving vehicles cannot effectively and timely identify all the factors that may affect driving safety.

The camera system on AVs depends on the brightness of the scene to determine the intensity of image pixels [66]. Since night vision images have fewer texture details and low contrast, dim light in the evening would reduce the ability of AVs to recognize the surrounding scene and lack sufficient reaction time to avoid serious crashes [67,68]. Moreover, rain may obscure the edges of objects, making them difficult to recognizable. Although radar may not be affected by dark conditions or rain, vulnerable road users cannot be identified accurately [69]. Human drivers also have poor visualization at night, however, most of them would be more vigilant, which may lead to relatively low speeds to avoid serious crashes. Visibility may provide safety improvement with the advance of machine vision and digital image processing techniques. In addition, roadway, intersection, and personal lighting, reflective materials, night vision, and educational interventions are also important [70].

#### 4.1.2. Road Variables

The existence of bicycle lanes increased the likelihood of the high crash severity compared to the road with only motor lanes in both driving modes. The likelihood increased by 22% in the autonomous driving mode (OR = 1.22), while in the conventional mode, there was an increase of 42% (OR = 1.42), representing a greater positive impact (See Table 3, Table 4, and Figure 5b). The existence of bicycle lanes means there are more non-motor vehicles in these sections. Compared with the collision between vehicles, non-motor vehicles are more frequently injured in crashes. AVs’ technologically advanced sensors and algorithms, and the potential ability of bicycles to communicate with AVs via transponders, are viewed as reasons for greater ability to perceive cyclists [71]. Thus, human drivers are more vulnerable to the emergence of bicycles on the road.

As shown in Table 3 and Figure 3, compared to no crosswalk roadway or road with only one side crosswalk, the likelihood of high crash severity was increased at roadways with crosswalks on both sides by 27% in the autonomous driving mode (OR = 1.27). Lots of research about the interaction between pedestrians and AVs have shown that the ability of AVs to detect and understand responses from pedestrians and respond appropriately is not yet complete [72,73]. Without pedestrian-to-driver communication (e.g., eye contact), pedestrian behavior becomes more unpredictable [74].

As for the conventional driving mode, compared with the number of lanes at the crash site less than 2, the possibility of high crash severity raised by 19% (OR = 1.19) if the number of lanes was more than 2 (See Table 4 and Figure 4). As with previous studies, more lanes are positively related to higher crash severity, since they commonly mean higher speed, which may cause higher crash severity [75,76]. The advantages of AVs, such as avoiding driver errors and better deceleration performance, make them perform better than human drivers in the multi-lane scenario.

#### 4.1.3. Vehicle Variables

The state of the vehicle impacted the likelihood of the high crash severity in both driving modes. When compared to stopped vehicles, the possibility of the high crash severity rose by 25% with moving vehicles in the conventional mode (OR = 1.25). In the autonomous mode, there was a 57% increase (OR = 1.57) (See Table 3, Table 4, and Figure 5c). From the perspective of kinetic energy, it is easy to explain that moving vehicles would cause more serious consequences. However, the different degrees of increase in the two modes are worthier attention. Previous studies have reported that AVs have a significant impact on the uncertainty, conflict, and stability of mixed traffic, which are highly associated with the severity of crashes [77,78]. Although the safety performance could substantially improve with a high penetration of AVs, AVs would adversely affect the traffic environment at lower penetration [79]. Moreover, drivers may game with the limitations of AVs and behave more aggressively in their vicinity [80]. Therefore, popularizing AV knowledge for conventional drivers and improving their vigilance in mixed traffic may help reduce crash severity. While studying car-following in mixed traffic, the interaction between human drivers and AVs should be paid more attention [81].

As shown in Table 4 and Figure 4, compared with vehicles operating straight ahead, the likelihood of the high crash severity rose by 22% (OR = 1.22) when the vehicle was making a turning motion in the conventional mode. Turning movements are usually associated with much higher attention than normal moving [82]. AVs with powerful perception and decision-making systems may perform better than human drivers.

### 4.2. Model Comparison

To examine whether the hierarchical Bayesian models with random intercept and random slopes are superior, this study made a comparison among models with different structures. Overall, models in both driving modes can be grouped into the following categories: (a) Bayesian logistic regression models (with only fixed effects) (b) hierarchical Bayesian models with random intercept (c) hierarchical Bayesian models with both random intercept and random slopes.

The lower the WAIC or LOO value, the better the model. As shown in Table 5, hierarchical Bayesian models with both random intercept and random slopes performed better than the other two kinds of models with lower WAIC and LOO.

In addition, the differences between models with different observation units were also explored. As shown in Figure 6a, in this study, vehicles with the same vehicle year from the same company (i.e., the “Vehicle company & year” unit) were considered as a cluster (Level 2) to alleviate the effects of unobserved heterogeneity, and there were several sub-clusters per cluster, i.e., each crash (level 1). To consider the possible interconnect role of collision manner in the relationship between independent variables and crash severity, “Crash type” was taken as the criterion for classifying clusters (See Figure 6b). Since studies have shown that robust parameter estimation cannot be made in the hierarchical Bayesian model if clusters in level 2 were too little [83], “Crash type” was not a dichotomous variable here, but was classified into 6 groups, including “Rear-end”, “Sideswipe”, “Head-on”, “Hit pedestrian”, “Hit non-motor vehicle”, and “Others”. Moreover, while considering the unobserved heterogeneities caused by both “Vehicle company & year” and “Crash type” (See Figure 6c), another hierarchical Bayesian model with two clusters was also attempted.

As shown in Table 6, the 2-level hierarchical Bayesian models with “Vehicle company & year” unit as level 2 had the lowest WAIC and LOO values, indicating that it performed better than the other two kinds of models.

### 4.3. Practical Research Implications

The findings of this study have several practical implications:(1)This study can help to improve driving safety in both conventional and autonomous driving modes. AVs are a complex combination of various hardware and software with high costs. Strict laws and regulations should be formulated to specify the road section and time of AVs for testing and driving [5]. For example, the testing of AVs needs to be conducted in areas with a complex environment or at night. In addition, manufacturers should face up to the current situation that AVs are still unable to effectively deal with all factors affecting driving safety and eliminate exaggerated publicity. More importantly, despite the importance of economic benefit and efficiency for manufacturers, the improvement of the safety of AVs should not be ignored. Different influencing factors for crashes mean different perceptions and decision logic. When AVs and conventional vehicles are mixed on the road, the driving environment may be more complex, and the risk of crashes may increase. Improving the technology of autonomous driving will help to reduce the collision between them. Additionally, as for the conventional driving mode, drivers’ behavior needs to be attached more importance. In a scene with more pedestrians or complex traffic flow, crashes in the conventional mode will be decreased by setting traffic signs and lines, violation penalties, and other means [84,85];(2)This study shows some existing problems of autonomous driving vehicles, which are helpful for the intelligent transformation of highways and vehicles. With the popularity of the Internet of Things (IoT), vehicle information and all kinds of environmental data can be collected in real-time through the wide application of artificial intelligence and big data [86,87]. At present, the intellectualization of highways should focus more on complex road environments and solving the occlusion problem, so as to provide clearer information for vehicles. Learning ability and adaptability need to be improved for AVs, enabling them to become a moving “intelligent agent”. Each “intelligent agent” can coordinate their respective routes, speeds, and distances between other vehicles to independently cope with all kinds of road conditions and unexpected situations [88,89]. In addition, the integration of multiple perception systems, such as visual perception systems (cameras and visual sensors), laser perception systems (laser radar), and microwave perception systems (millimeter-wave radar), will also help to reduce autonomous driving crashes;(3)The content of this study has a certain reference value for the research of human-computer interaction. Semi-AVs will occupy the majority of the market for a long time in the future, however, it is difficult to define when the driver or vehicle should be responsible for driving. At present, AVs require human drivers to maintain control of vehicles, during the whole driving process. However, when an emergency occurs, drivers may not be able to take charge of vehicles immediately because of carelessness or gradual trust in the autonomous driving system. The warning systems shall be further improved for AVs. The core of human-computer interaction is coordination and complementarity [90,91]. Since autonomous driving technology is not yet fully mature, semi-autonomous driving can be used as a supplement to traditional driving to reduce crashes. The human-computer conflict caused by redundant input also needs to be avoided [92,93].

## 5. Conclusions

This study aims to analyze the divergent influences of factors on crashes under the autonomous driving mode and the conventional driving mode. By using the hierarchical Bayesian approach to consider unobserved heterogeneities, crash severity was analyzed. The results showed that there were significant differences in the severity of influencing factors under different driving modes.

Although some influencing factors had the same positive or negative effects on crash severity under both driving modes, their degrees were different. The impact of daily visitors’ flowrate (DVF) on crash severity in the autonomous driving mode was smaller than that in the conventional mode since equipped with advanced sensing equipment, AVs can sense a longer distance and were superior to humans in the recognition of specific targets (e.g., face, text, etc.) [94]. In addition, the presence of bike lanes would lead to a great increase in the severity of crashes in the conventional driving model, while the moving vehicle state had a greater impact on the crash severity of autonomous driving.

More influencing factors only had a significant impact on one of the driving modes, which was also worthy of analysis. To be specific, raining, mixed land use, the presence of the muni line, and driving at night would cause high injury severity in the autonomous mode, but in the conventional driving mode, their impacts were not significant. The problems for autonomous driving vehicles in the total factor recognition and completion of vision occlusion in complex environments may lead to this result. Furthermore, more lanes would increase the possibility of high crash severity in the conventional mode because of human drivers’ wrong decisions, but AVs can effectively avoid that.

There are some limitations in this study that should be addressed in future work. Although the Traffic Collision Involving an Autonomous Vehicle Reports (OL 316) contain much useful information, some important data, such as traffic flow, pre-crash vehicle kinematic data, driver demographics, driver perception, etc., were not recorded in the report. This study used Bayesian inference to reduce the influence of a small sample, but more crash reports will of course lead to more profound and generalizable insights. Additionally, sufficient crash data makes it possible to consider more potential relationships, such as cross-level interactions of impacting factors. Moreover, it should be noted that the AV’s behavior may be timid in the test stage on public roads compared to their future behavior. With the maturity and commercialization of AVs throughout the world, an updated evaluation will be required. With the AV crash data constantly updated, we will continue to add new crashes and more features to make a more detailed classification of the crash severity and improve the accuracy and effectiveness of our models in future research.

## Figures and Tables

**Figure 1 ijerph-19-11358-f001:**
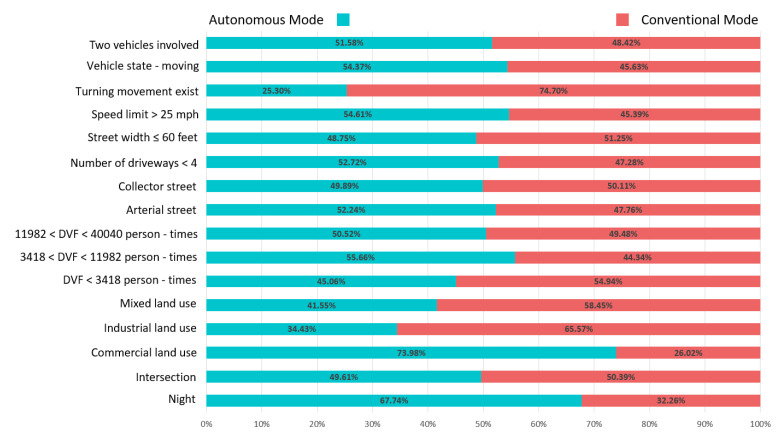
A comparison of driving conditions in both driving modes.

**Figure 2 ijerph-19-11358-f002:**
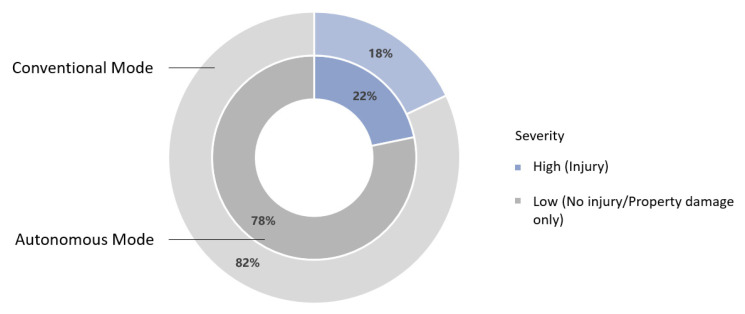
Statistical results of crash severity in both driving modes.

**Figure 3 ijerph-19-11358-f003:**
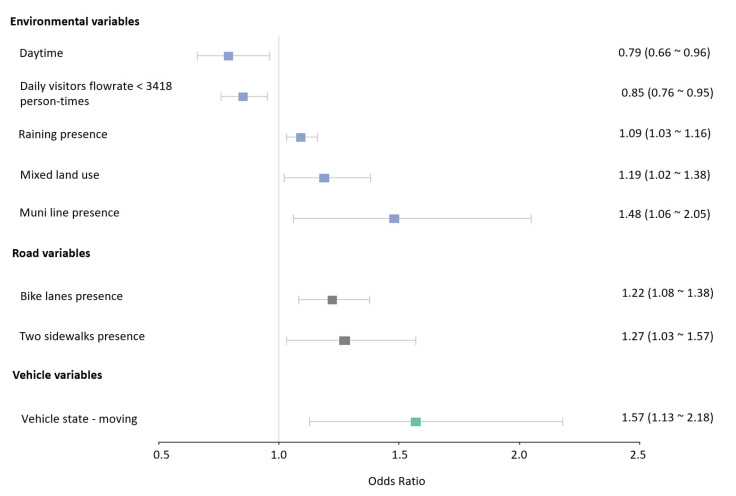
Odds Ratio of the influencing factors for crash severity in the autonomous mode.

**Figure 4 ijerph-19-11358-f004:**
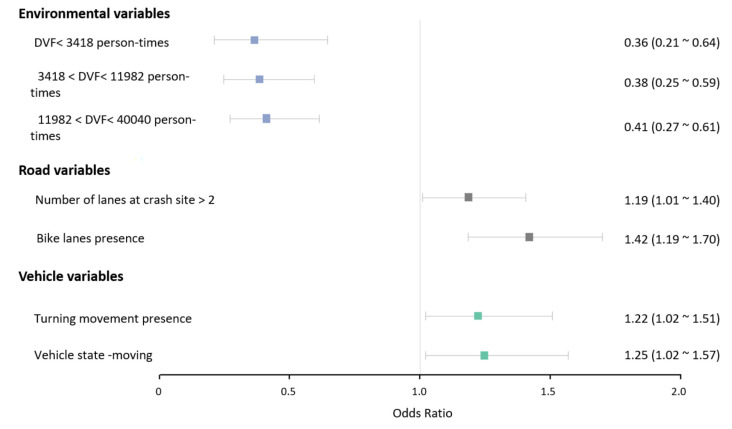
Odds Ratio of the influencing factors for crash severity in the conventional mode.

**Figure 5 ijerph-19-11358-f005:**
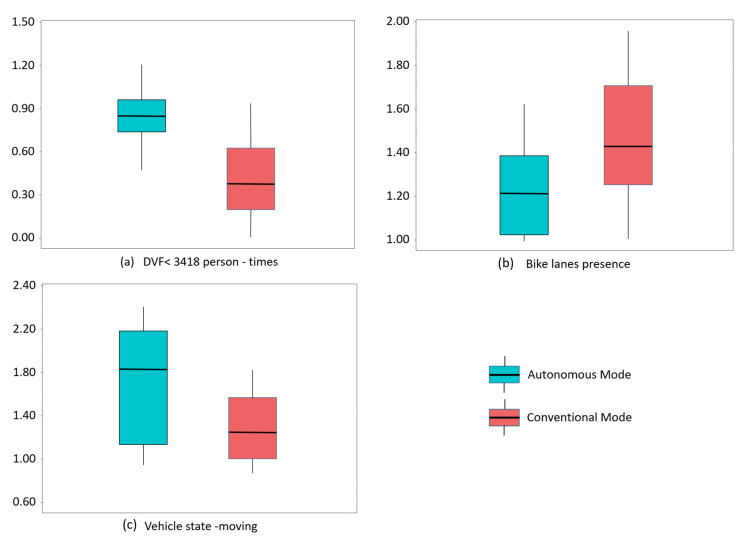
Comparison of the same influencing factors for crash severity in autonomous and conventional driving modes.

**Figure 6 ijerph-19-11358-f006:**
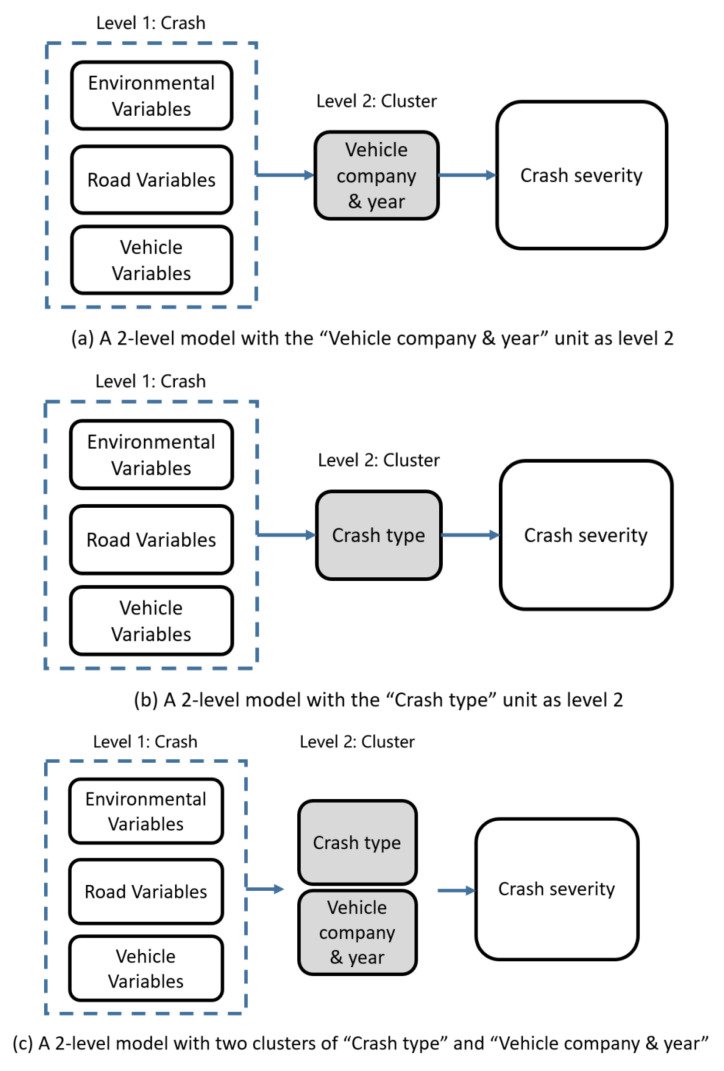
The structure of hierarchical Bayesian model.

**Table 1 ijerph-19-11358-t001:** Descriptive statistics of continuous variables.

Variable Category	Description	Mean	S.D.	Min	Max
Environmental Variables					
Schools	Count of public and private schools within a quarter-mile	1.94805	1.8499	0	9
Parks	Count of parks within a quarter-mile	1.89286	1.46636	0	6
Road Variables					
Driveway	Count of driveways along segment	3.12987	1.36579	1	8
Crash lanes	Number of lanes at crash site	2.12338	1.07453	1	6
Street width	Width of street in feet	51.85065	19.63816	22	140
Speed limit	Speed limit of roadway in mph	25.42208	1.71034	15	30
Slope	Slope in percentage of roadway	3.41558	2.95931	1	10

**Table 2 ijerph-19-11358-t002:** Descriptions, distribution, and sources of explanatory variables.

Variable Category	Description	Variable	Autonomous		Conventional		Source
			Num	Percent	Num	Percent	
Injury	Someone injured	No *	74	77.08%	70	83.33%	OL 316
		Yes	22	22.92%	14	16.67%	
Environmental Variables							
Time of day	Time of the crash	Daytime	60	62.50%	69	82.14%	OL 316
		Night *	36	37.50%	15	17.86%	
Involved in the crash	Non-motor vehicles or pedestrians involved in the crash	No	78	81.25%	58	69.05%	OL 316
		Yes *	18	18.75%	26	30.95%	
Intersection	Crash happened at an intersection	No *	33	34.38%	28	33.33%	OL 316
		Yes	63	65.63%	56	66.67%	
Light	Presence of light	Dark *	54	56.25%	5	5.95%	OL 316
		Daylight	42	43.75%	79	94.05%	
Roadway surface	Condition of roadway surface	Dry	91	94.79%	75	89.29%	OL 316
		Wet *	3	3.13%	6	7.14%	
		Unknown	2	2.08%	3	3.57%	
Metro stop	Presence of metro stop	Absence *	51	53.13%	39	46.43%	TransBASE
		Presence	45	46.88%	45	53.57%	
Trees	Presence of trees	Absence *	19	19.79%	23	27.38%	TransBASE
		Presence	77	80.21%	61	72.62%	
Land use	Land use of the location	Commercial	26	27.08%	8	9.52%	TransBASE
		Industrial	3	3.13%	5	5.95%	
		Mixed or public	39	40.63%	48	57.14%	
		Residential *	28	29.17%	23	27.38%	
Weather	Weather at the time of the crash	Clear weather *	85	88.54%	74	88.10%	OL 316
		Cloudy	5	5.21%	7	8.33%	
		Fog/Visibility	2	2.08%	0	0.00%	
		Raining	3	3.13%	3	3.57%	
		Unknown	1	1.04%	0	0.00%	
Muni line	Presence of muni line (i.e., public transport line)	Absence *	20	20.83%	12	14.29%	TransBASE
		Presence	76	79.17%	72	85.71%	
Daily visitors’ flowrate (DVF)	Level of DVF	DVF < 3418 person-times	30	31.25%	32	38.10%	TransBASE
		3418 person-times ≤ DVF < 11,982 person-times	33	34.38%	23	27.38%	
		11,982 person-times ≤ DVF < 40,040 person-times	28	29.17%	24	28.57%	
		DVF ≥ 40,040 person-times *	5	5.21%	5	5.95%	
Pavement markings conditions	conditions of pavement markings	Poor *	6	6.25%	6	7.14%	Google Earth
		Adequate	90	93.75%	78	92.86%	
Schools	Count of public and private schools within a quarter-mile	Count of schools > 4	20	20.83%	16	19.05%	TransBASE
		Count of schools ≤ 4 *	76	79.17%	68	80.95%	
Parks	Count of parks within a quarter-mile	Count of parks > 4	6	6.25%	5	5.95%	TransBASE
		Count of parks ≤ 4 *	90	93.75%	79	94.05%	
Road Variables							
Street classification	Classification of street	High	1	1.04%	0	0.00%	TransBASE
		Arterial	20	20.83%	16	19.05%	
		Collector	33	34.38%	29	34.52%	
		Residential *	42	43.75%	39	46.43%	
One-way	One-way street	No *	62	64.58%	56	66.67%	TransBASE
		Yes	34	35.42%	28	33.33%	
Divided median	Presence of divided median	Absence *	80	83.33%	76	90.48%	TransBASE
		Presence	16	16.67%	8	9.52%	
Marked centerline	Presence of marked centerline	Absence *	56	58.33%	43	51.19%	TransBASE
		Presence	40	41.67%	41	48.81%	
Bike lane	Presence of bike lane	Absence *	70	72.92%	54	64.29%	TransBASE
		Presence	26	27.08%	30	35.71%	
On-street parking	Presence of on-street parking	Absence *	15	15.63%	11	13.10%	TransBASE
		Presence	81	84.38%	73	86.90%	
Off-street parking	Presence of off-street parking	Absence *	1	1.04%	3	3.57%	TransBASE
		Presence	95	98.96%	81	96.43%	
Traffic calming	Presence of traffic calming device	Absence *	69	71.88%	58	69.05%	TransBASE
		Presence	27	28.13%	26	30.95%	
Sidewalk	Presence of sidewalk	Absence or one-side of segment *	5	5.21%	7	8.33%	TransBASE
		Both sides of segment	91	94.79%	77	91.67%	
Driveway	Count of driveways along segment	Driveways ≥ 4 *	31	32.29%	33	39.29%	TransBASE
		Driveways < 4	65	67.71%	51	60.71%	
Crash lanes	Number of lanes at crash site	Crash lanes > 2	36	37.50%	27	32.14%	TransBASE
		Crash lanes ≤ 2 *	60	62.50%	57	67.86%	
Street width	Width of street in feet	Street width > 60 feet	21	21.88%	15	17.86%	Google Earth
		Street width ≤ 60 feet *	75	78.13%	69	82.14%	
Speed limit	Speed limit of roadway in mph	Speed limit > 25 mph	11	11.46%	8	9.52%	TransBASE
		Speed limit ≤ 25 mph *	85	88.54%	76	90.48%	
Slope	Slope in percentage of roadway	Slope > 3%	42	43.75%	31	36.90%	TransBASE
		Slope ≤ 3% *	54	56.25%	53	63.10%	
Vehicle Variables							
Turning movement	Turning movement of the AV	No *	84	87.50%	53	63.10%	OL 316
		Yes	12	12.50%	31	36.90%	
Manufacturer	Manufacturer of the AV	Aurora Innovation, Inc. (Pittsburgh, PA, USA)	0	0.00%	1	1.19%	OL 316
		GM Cruise LLC (San Francisco, CA, USA)	79	82.29%	53	63.10%	
		Lyft, Inc. (San Francisco, CA, USA)	0	0.00%	2	2.38%	
		Waymo LLC (Phoenix, AZ, USA)	8	8.33%	9	10.71%	
		Zoox, Lnc. (San Francisco, CA, USA)	9	9.38%	19	22.62%	
Vehicle year	Production year of the AV	2016	9	9.38%	17	20.24%	OL 316
		2017	20	20.83%	16	19.05%	
		2018	0	0.00%	1	1.19%	
		2019	21	21.88%	15	17.86%	
		2020	45	46.88%	34	40.48%	
		2021	1	1.04%	1	1.19%	
Vehicle state	State of AV	Stopped *	32	33.33%	37	44.05%	OL 316
		Moving	64	66.67%	47	55.95%	
Crash type	Type of the crash Rear-end	Rear-end	57	59.38%	34	40.48%	OL 316
		Other *	39	40.63%	50	59.52%	
Number of vehicles involved	Number of vehicles involved in the crash	1 *	11	11.46%	13	15.48%	OL 316
		2	84	87.50%	69	82.14%	
		3	1	1.04%	2	2.38%	
Disengagement	Presence of disengagement	Absence *	60	62.50%	84	100.00%	OL 316 &OL311R
		Presence	36	37.50%	0	0.00%	
Initiator of disengagement	Initiator of disengagement (system or the test driver)	AV system	1	1.04%	0	0.00%	OL 316 &OL311R
		Test driver	35	36.46%	0	0.00%	
		No	60	62.50%	84	100.00%	
Unwanted behavior of other roadway participants	Presence of unwanted behavior of other roadway participants	Absence *	77	80.21%	84	100.00%	OL 316 &OL311R
		Presence	19	19.79%	0	0.00%	
Unwanted movement of AVs	Presence of unwanted behavior of AVs	Absence *	95	98.96%	84	100.00%	OL 316 &OL311R
		Presence	1	1.04%	0	0.00%	
Changing lanes	Presence of AV’s changing lanes	Absence *	64	66.67%	84	100.00%	OL 316 &OL311R
		Presence	32	33.33%	0	0.00%	
Deceleration	Presence of AV’s deceleration	Absence *	76	79.17%	84	100.00%	OL 316 &OL311R
		Presence	20	20.83%	0	0.00%	

* denotes the reference group.

**Table 3 ijerph-19-11358-t003:** The hierarchical Bayesian model for crash severity in the autonomous mode.

Parameters	Estimate (std Error)	Odds Ratio (95% Confidence Interval)
Fixed Effects		
Environmental variables		
Daytime	−0.23 (0.08)	0.79 (0.66~0.96)
Night *	0	1
Daily visitors’ flowrate (DVF) < 3418 person-times	−0.16 (0.10)	0.85 (0.76~0.95)
DVF > 40,040 person-times *	0	1
Raining presence	0.09 (0.27)	1.09 (1.03~1.16)
Raining absence *	0	1
Mixed land use ^#^	0.17 (0.12)	1.19 (1.02~1.38)
Residential land use *	0	1
Muni line presence	0.39 (0.09)	1.48 (1.06~2.05)
Muni line absence *	0	1
Road variables		
Bike lanes presence	0.20 (0.09)	1.22 (1.08~1.38)
Bike lanes absence *	0	1
Two sidewalks presence	0.24 (0.17)	1.27 (1.03~1.57)
Absence or only one sidewalk *	0	1
Vehicle variables		
Vehicle state-moving	0.45 (0.28)	1.57 (1.13~2.18)
Vehicle state-stopped *	0	1
Intercept (level 1)	0.45 (0.32)	1.57 (1.01~2.44)
Random effects		
Vehicle state-moving	0.16 (0.13)	1.17 (1.00~1.38)
Intercept (Vehicle company & year)	0.09 (0.10)	1.09 (1.06~1.13)
WAIC	62.8	
LOO	63.4	

* denotes the reference group; ^#^ denotes the random variable.

**Table 4 ijerph-19-11358-t004:** The hierarchical Bayesian model for crash severity in the conventional mode.

Parameters	Estimate (std Error)	Odds Ratio (95% Confidence Interval)
Fixed effects		
Environmental variables		
Daily visitors’ flowrate (DVF) < 3418 person-times ^#^	−1.01 (0.28)	0.36 (0.21~0.64)
3418 < DVF < 11,982 person-times	−0.96 (0.22)	0.38 (0.25~0.59)
11,982 < DVF < 40,040 person-times	−0.89 (0.21)	0.41 (0.27~0.61)
DVF > 40040 person-times *	0	1
Road variables		
Number of lanes at crash site > 2	0.17 (0.10)	1.19 (1.01~1.40)
Number of lanes at crash site ≤ 2 *	0	1
Bike lanes presence	0.35 (0.09)	1.42 (1.19~1.70)
Bike lanes absence *	0	1
Vehicle variables		
Turning movement presence	0.20 (0.10)	1.22 (1.02~1.51)
Turning movement absence *	0	1
Vehicle state-moving	0.22 (0.11)	1.25 (1.02~1.57)
Vehicle state-stopped *	0	1
Intercept (level 1)	0.74 (0.23)	2.09 (1.32~3.19)
Random effects		
DVF < 3418 person-times	0.30 (0.23)	1.35 (1.03~2.53)
Intercept (Vehicle company & year)	0.21 (0.28)	1.23 (1.01~3.63)
WAIC	52.6	
LOO	53.3	

* denotes the reference group; ^#^ denotes the random variable.

**Table 5 ijerph-19-11358-t005:** WAIC and LOO of hierarchical Bayesian models with different structures.

	Bayesian Logistic Regression Models (with Only Fixed Effects)		Hierarchical Bayesian Models with Random Intercept		Hierarchical Bayesian Models with Both Random Intercept and Random Slopes	
	WAIC	LOO	WAIC	LOO	WAIC	LOO
Models for crash severity in the autonomous mode	74.4	76.5	64.5	64.9	62.8	63.4
Models for crash severity in the conventional mode	60.9	61.9	53.3	53.7	52.6	53.3

**Table 6 ijerph-19-11358-t006:** WAIC and LOO of hierarchical Bayesian models with different observation units.

	The 2-Level Hierarchical Bayesian Models with “Vehicle Company & Year” Unit as Level 2		The 2-Level Hierarchical Bayesian Models with “Crash Type” Unit as Level 2		A 2-Level Model with Two Clusters of “Crash Type” and “Vehicle Company & Year”	
	WAIC	LOO	WAIC	LOO	WAIC	LOO
Models for crash severity in the autonomous mode	62.8	63.4	65.1	66.0	63.1	63.8
Models for crash severity in the conventional mode	52.6	53.3	54.7	55.2	53.5	54.0

## Data Availability

The datasets analyzed for this study can be found in the https://www.dmv.ca.gov/portal/vehicle-industry-services/autonomous-vehicles/autonomous-vehicle-collision-reports/ (accessed on 1 April 2021); https://www.dmv.ca.gov/portal/dmv/detail/vr/autonomous/testing (accessed on 1 April 2021); http://transbasesf.org/transbase/ (accessed on 1 April 2021); https://www.google.com/earth (accessed on 1 April 2021).

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
