# Peer review of "Divergent Effects of Factors on Crash Severity under Autonomous and Conventional Driving Modes Using a Hierarchical Bayesian Approach"

_ijerph, 2022, doi:10.3390/ijerph191811358_

Round 1
Reviewer 1 Report
The paper shows a kind of statistical model of AV's crash severity for accident analysis. Most of the methods are resemble to [30]. But the model superiority is not clearly shown.
pp.2,L84.
Please evaluate the effectiveness of your hierarchical model. First flat model should be established and compared with hierarchical ones. Your approach really solved this unobserved heterogeneity problem?
The research method looks very similar to [30]. Please clarify your contribution comparing with [30].
pp.4,L168
What is muni line?
pp.4,L188-190
Does not look similar. For example, commercial land use and Night looks higher in autonomous mode.
pp.5, L204
41 variables, but 40 variable in Table 2?
Data source is the same as [30], but variables are not; some are added, and some are omitted. Why you need to changed the variables used in [30]?
pp.8,L219-222
What variables are VIF > 10 and eliminated?
pp.8-L252
Eq(1) is not probability but logit model.
pp.10 L297
What are sigma_0 and sigma_k?
pp.11 L334-335
Please clarify 8 variables for autonomous mode and 7 variables for conventional mode. Table.3, 4 includes much more variables. Fig.3 and Fig.4 may be. How are these variables selected?
pp.14 L390-391
Where is the evidence? The autonomous driving mode always exhibits higher odds ration for DVFs than conventional.
pp.17 Fig.6 (c)
Where is the third level hierarchical formulation? In 2.2.1, only second level formulation is showed.
pp.9 eq.(4)
Your hierarchical Bayesian model looks different from generic Hierarchical Bayesian model. Generally, hierarchical Bayesian model is like this.
Level 0: P(z|y,x)
Level 1: P(y|x)
Level 2: P(x)
z: crash severity
y: vehicle company & year
x: environment, road, vehicle
Eq.(4) looks non-linear logistic model. The variables X and Z does not form Bayesian chain. What is your definition of hierarchical Bayesian model.
Reviewer 2 Report
Dear Authors,
the manuscript presented for evaluation concerns a very current topic of road safety with the use of autonomous and traditional vehicles using the Hierarchical Bayesian Approach to evaluation. The authors quite comprehensively presented their research and the obtained results using well-selected literature, although it is possible to supplement it with a few recent items. The manuscript is well prepared, it includes all the necessary chapters, and the analysis of the results leads to the right conclusions. You can see a lot of experience and the authors' care for the substantive level, unfortunately there are some minor errors that should be corrected before publishing the article.
Detailed notes on the manuscript:
Firstly, I have comments on the literature template used, please check before submitting your final manuscript.
I suggest adding some literature on the safe use of autonomous vehicles (line 34), these studies may be helpful:
10.1038/s41562-017-0202-6
https://doi.org/10.1515/eng-2021-0087
10.20858 / sjsutst.2018.100.2
10.1016 / j.trpro.2020.02.031
https://doi.org/10.3390/en14185778
For better readability, the following drawings can be enlarged: Fig. 1, Fig. 6.
It is worth considering moving a part of the text from the Conclusions chapter to the Discussion chapter under Table 6. I am referring to an excerpt from verses 527-568. I believe that the Conclusions section is too long, and a sub-section 4.3. "Practical Research Implications" could be added.
Thank you
Round 2
Reviewer 1 Report
I've not been convinced of the model yet.
Eq(4), in your reply, nonlinear terms XZ exist. Thus I thought that it is not a linear model. How did you fitted the model using R?
See below.
The model brm package assumes is purely linear as
X.b+ Z.u
Reviewer 2 Report
Dear Authors,
thank you for your responses and for taking into account my suggestions for the manuscript. I have read the text and the Authors' answers. I am satisfied with the submitted version of the manuscript, I accept the comments and the changes made in the text. I believe that the article may be submitted to print in its current form. Thank you for being able to see your work.
I wish you further scientific success.
Author Response
We would like to thank you for your recommendation for publication and thoughtful suggestions that have helped to improve this paper substantially. Besides, thank you for your affirmation of our study.